The cuproptosis-related gene ITGB6 and LTBP1 may be associated with diabetic kidney disease progression and immune cell infiltration

Hu Suying 1 2
Tian Mengdi 2
Hu Wenjia 2
Yao Liang 2
Tang Ying 1
Shen Wei 1
He Qing 1
Xu Jing 1
Yao Huan 1
Ji Lei 3
Fan Feifei 4
Liu Shiqiang 4
Wang Zhen 5 wangzhen170111@dingtalk.com
1 Department of Endocrinology, Wuhu Hospital of Traditional Chinese Medicine , Wuhu , China
2 Wuhu Hospital of Traditional Chinese Medicine, Anhui University of Chinese Medicine Postgraduate Training Base , Wuhu , China
3 Department of Anesthesiology, Wannan Medical College , Wuhu , China
4 Department of Endocrinology, First Affiliated Hospital of Wannan Medical College , Wuhu , China
5 Department of Cardiovascular Surgery, Xi’an International Medical Center Hospital Affiliated to Northwest University , Xi’an , China
Nakai Kenta
Electronic publication date: 2025 Nov 11
Publication date: 2025
Volume: 13
Electronic Location ID: e20346
Received 2025 Jun 17; Accepted 2025 Oct 15
Copyright: © 2025 Hu et al.
Copyright year: 2025
Copyright holder: Hu et al.
License: This is an open access article distributed under the terms of the Creative Commons Attribution License, which permits unrestricted use, distribution, reproduction and adaptation in any medium and for any purpose provided that it is properly attributed. For attribution, the original author(s), title, publication source (PeerJ) and either DOI or URL of the article must be cited.
License URL: https://creativecommons.org/licenses/by/4.0/

Keywords: Cuproptosis, ITGB6, LTBP1, MPC5, Diabetic kidney disease

Funding: Key Project of Wuhu Health Commission WHWJ2023z002 This work was supported by the Key Project of Wuhu Health Commission (“Mechanism of novel cell death mode ‘Cuproptosis’ in diabetic kidney disease”; Grant No. WHWJ2023z002). The funders had no role in study design, data collection and analysis, decision to publish, or preparation of the manuscript.

==============================
Objective

Cuproptosis, a newly discovered cell death mechanism, has been linked to the pathogenesis of multiple diseases. However, its role in diabetic kidney disease (DKD) remains unclear.

Methods

By analyzing datasets GPL17586 and GPL571 from the GEO database and applying machine learning, cuproptosis-related marker genes associated with DKD were identified. The expression levels of these genes were examined in Mouse Podocyte Cell Line (MPC5) podocytes cultured in vitro and treated with high glucose (30 mM) for 24, 48, and 72 h to explore their roles in the onset and progression of DKD.

Results

Key genes in the cuproptosis pathway, integrin β6 (ITGB6) and latent transforming growth factor beta-binding protein 1 (LTBP1), were significantly upregulated in DKD patients. Consistent with this, in high glucose-treated podocytes, the expression of ITGB6 and LTBP1 was significantly higher than in the control group at 24, 48, and 72 h. The area under the receiver operating characteristic (ROC) curve (AUC) for ITGB6 and LTBP1 in both the training set (GPL17586) and validation set (GPL571) exceeded 0.7, indicating good diagnostic efficacy for DKD. Furthermore, immune infiltration analysis further revealed that ITGB6 and LTBP1 were significantly positively correlated with activated B cells, central memory Cluster of Differentiation 4 (CD4) T cells, effector memory CD4 T cells, effector memory Cluster of Differentiation 8 (CD8) T cells, and immature B cells, while showing a significant negative correlation with neutrophils.

Conclusion

This study suggests that cuproptosis-related genes ITGB6 and LTBP1 may be associated with the progression of DKD through their potential role in immune cell infiltration, and could serve as potential novel targets for the prevention and diagnosis of DKD.

Introduction

Diabetic kidney disease (DKD) is one of the major microvascular complications of diabetes and remains the leading cause of end-stage renal disease (ESRD) worldwide (Zhang et al., 2021). Currently, approximately 20% to 40% of diabetic patients develop DKD (de Boer et al., 2011; Afkarian et al., 2016), and in patients with type 2 diabetes mellitus (T2DM) in China, the prevalence of DKD is 21.8% (Zhang, Kong & Yun, 2020). Although previous studies have revealed the association of DKD with pathological processes such as inflammation, fibrosis, and oxidative stress, these mechanisms still fail to fully explain the occurrence and progression of DKD, particularly the significant individual variability observed among patients (Ahmad, Draves & Rosca, 2021; Chen, Min & Wang, 2022; Watanabe et al., 2022).

Copper is one of the essential trace elements in organisms, participating in various important physiological functions. The balance of copper metabolism is primarily regulated by a series of proteins and enzymes, such as copper transport P-type ATPases and copper chaperones (Tsvetkov et al., 2022). However, when copper metabolism is imbalanced, the accumulation of copper ions can trigger a novel form of cell death known as copper toxicity (cuproptosis) (Chen, Min & Wang, 2022; Xue et al., 2023). From a pathological perspective, excess copper can induce granular degeneration, vacuolation, and necrosis of renal tubular epithelial cells, particularly in the proximal tubules, ultimately leading to acute or chronic renal failure (Wong et al., 2020; Gujarati et al., 2021).

Cuproptosis interferes with normal mitochondrial function, thereby inducing cell death, a process that has been shown to be closely related to the development of various diseases (Baraka-Vidot et al., 2014; Chen et al., 2023a; Zhong et al., 2024). Clinical evidence indicates that copper metabolism disorder is common in DKD patients, typically characterized by elevated serum copper levels and increased urinary copper excretion (Yang et al., 2019). These alterations not only reflect systemic metabolic abnormalities but are also closely associated with the progression of renal dysfunction and structural kidney damage in DKD (Gembillo et al., 2022). A Mendelian randomization study further supports the role of copper in kidney disease, demonstrating that elevated circulating copper levels are significantly correlated with chronic kidney disease, renal injury, and decline in renal function (Ahmad, Ärnlöv & Larsson, 2022).

Moreover, hypoxia-reoxygenation during renal ischemia-reperfusion injury leads to Fe(II) accumulation and downregulation of [4Fe-4S] cluster assembly proteins, resulting in structural disruption of lipoic acid synthetase (LIAS) and lipoylation deficiency, thereby triggering cuproptosis. While these findings highlight the possibility of cuproptosis occurring in renal tissues under stress conditions, its specific role in diabetic nephropathy—particularly in key cell types such as podocytes—has not been directly investigated (Chen et al., 2025).

Given the emerging role of cuproptosis in human diseases, this study explored its potential involvement in DKD. Through bioinformatics analyses of public datasets and in vitro validation, we identified integrin β6 (ITGB6) and latent transforming growth factor beta-binding protein 1 (LTBP1) as cuproptosis-related genes dysregulated in DKD. Their associations with immune cell infiltration further suggest a possible link between cuproptosis, immune dysregulation, and DKD progression, providing new insights into potential diagnostic and therapeutic targets.

Materials and Methods

Data sources

Training set: The DKD dataset GPL17586 (Liu et al., 2018) was downloaded from the GEO database (https://www.ncbi.nlm.nih.gov/geo/), which utilizes chip sequencing (platform: GPL17586). The GPL17586 dataset was selected as the training set due to its comparatively larger sample size (40 DKD vs. 21 controls), well-annotated clinical phenotypes, and relevance to glomerular pathology in DKD. Kidney glomerular samples from 40 DKD patients were used as the DKD group, while glomerular samples from unaffected portions of the kidneys of 21 tumor nephrectomy patients were used as the control group.

Validation set: The validation set, GPL571 (Park et al., 2011), profiled on the GPL571 platform (Affymetrix Human Genome U133A 2.0 Array), was chosen as an independent cohort to evaluate the generalizability of our findings across different technical platforms. The GPL571 dataset was downloaded (platform: GPL571), including kidney glomerular samples from 9 DKD patients and 26 non-DKD patients as the DKD and control groups, respectively. Additionally, 36 cuproptosis-related genes were obtained from the literature (Chen et al., 2023b).

Data preprocessing subsection: To address potential batch effects between the GPL17586 and GPL571 datasets, which were generated on different microarray platforms (GPL17586 and GPL571, respectively), we applied the ComBat algorithm from the “sva” R package (version 3.46.0) to harmonize the expression data. The ComBat adjustment was performed using an empirical Bayes framework, with the dataset specified as the batch covariate and the disease group (DKD vs. control) preserved as a biological variable. This approach removed technical artifacts while retaining biologically relevant signals.

Differential expression analysis

The R programming language (version 4.3.2) and the Limma package (version 3.56.0) were used to identify differentially expressed genes (DEGs) between DKD and control groups in the GPL17586 dataset. Genes with an absolute log2fold change (|log2FC|) > 1 and an adjusted p-value (Benjamini-Hochberg method) < 0.05 were considered statistically significant. Visualization was performed using the R package ggplot2 (version 3.4.4) for volcano plots and ComplexHeatmap (version 2.16.0) for heatmaps.

ssGSEA analysis

Single-sample gene set enrichment analysis (ssGSEA), an extension of the gene set enrichment analysis (GSEA) method, was used to calculate enrichment scores for each sample based on the cuproptosis gene set. Wilcoxon rank-sum tests were applied to compare ssGSEA scores between the DKD and control groups, assessing the overall upregulation or downregulation of cuproptosis-related gene expression.

WGCNA analysis

Weighted gene co-expression network analysis (WGCNA) was performed using the WGCNA package (version 1.72-5) on gene expression data from 61 samples (DKD and control groups). A soft-thresholding power (β) of 5 was selected to ensure a scale-free topology (scale-free R2 > 0.85). Co-expression modules were identified using a dynamic tree-cutting algorithm (minModuleSize = 30, deepSplit = 2, mergeCutHeight = 0.25). Modules with a correlation P-value < 0.05 and |correlation coefficient| > 0.3 with the cuproptosis trait were considered significantly associated. Module membership (MM) > 0.8 and gene significance (GS) > 0.5 were used as cutoffs for identifying hub genes within key modules.

Machine learning for key gene selection

To identify candidate key genes with diagnostic potential, we employed two machine learning algorithms: eXtreme Gradient Boosting (XGBoost) and Least Absolute Shrinkage and Selection Operator (LASSO) regression.

For the XGBoost model, we used the “xgboost” package (version 1.7.5) in R. The hyperparameters were tuned via 10-fold cross-validation repeated 3 times to optimize generalization performance. The grid search space included: max_depth (3, 6), eta (0.01, 0.1), gamma (0, 0.2), subsample (0.7, 1), colsample_bytree (0.7, 1), and min_child_weight (1, 5). The final model was trained using the area under the receiver operating characteristics (ROC) curve (AUC) as the evaluation metric. Feature importance was assessed by the “gain” metric, which reflects the relative contribution of each gene to model accuracy.

For the LASSO regression, we utilized the “glmnet” package (version 4.1.7) in R. The optimal regularization parameter λ was selected through 10-fold cross-validation that minimized the binomial deviance. The value of λ corresponding to the minimum cross-validation error (λ.min) was chosen to identify non-zero coefficients and retain robust features. Genes with non-zero coefficients under the optimal λ were considered candidate diagnostic biomarkers. Both models were trained and validated using the GPL17586 and GPL571 datasets. The resulting key genes were then intersected from each algorithm to obtain a set of high-confidence diagnostic markers that reached a consensus.

Expression validation

The Wilcoxon test was used to analyze the expression differences of candidate key genes between the DKD and control samples (P < 0.05) in the GPL17586 training set and the GPL571 validation set. Boxplots were generated with ggplot2 to visualize the results. Genes showing significant differences and consistent trends (P < 0.05) in both datasets were considered key genes.

ROC curve analysis

ROC curve analysis was conducted to evaluate the diagnostic performance of candidate genes in distinguishing between DKD and control samples. Using the R package “pROC” (version 1.18.0), ROC curves were generated and the area under the curve (AUC) was calculated for each gene in both the training set (GPL17586) and the independent validation set (GPL571). Genes with an AUC value greater than 0.7 were considered to exhibit good diagnostic ability. The ROC curves visualize the trade-off between sensitivity and specificity across different threshold values, providing a quantitative measure of each gene’s potential as a diagnostic biomarker for DKD.

Nomogram construction

A nomogram was constructed based on a multivariate regression model integrating key genes with diagnostic value for DKD. The total score was calculated based on the weighted scores of each key gene, and the DKD probability was predicted based on the total score.

Enrichment analysis

Gene Ontology (GO) and Kyoto Encyclopedia of Genes and Genomes (KEGG) pathway enrichment analyses were performed for DE-hub genes using the R package “clusterProfiler.” GO analysis included biological processes (BP), cellular components (CC), and molecular functions (MF). KEGG pathway analysis was used to identify major metabolic and signaling pathways involving the differentially expressed genes.

PPI network analysis

A protein-protein interaction (PPI) network was constructed for DE-hub genes using the STRING database and visualized with Cytoscape software. Topological properties of the network (e.g., node degree and centrality) were analyzed to identify key proteins.

Immune cell infiltration analysis

To investigate the relationship between key genes and the immune microenvironment, ssGSEA was performed using the R package “GSVA” on the GPL17586 dataset to calculate immune cell enrichment scores. Wilcoxon tests were used to compare immune cell infiltration differences between the DKD and control groups, and Spearman correlation analysis was conducted to examine relationships between immune cells and diagnostic genes.

In vitro mouse podocyte culture and differential gene expression analysis

(1) Cell culture: The mouse renal podocyte cell line MPC5 was obtained from Procell Life Science & Technology Co., Ltd. (catalog number: CL-0855; RRID: CVCL_0496). Cells were cultured in RPMI-1640 medium (Gibco, Waltham, MA, USA) supplemented with 10% fetal bovine serum (FBS; Procell, Bethel, CT, USA) and 1% penicillin/streptomycin (Beyotime, Shanghai, China) at 37 °C in a humidified 5% CO2 atmosphere. High-glucose treatment was initiated when the cell density reached 80–90%. All cell experiments were conducted in accordance with relevant institutional guidelines and did not require additional ethical approval as the study exclusively used a commercially available cell line.

(2) High-glucose treatment: To study the effects of high glucose on LTBP1 and ITGB6 expression, MPC5 cells were divided into four groups: one control group (11.1 mM D-glucose concentration (Scrimieri et al., 2023)) and three high-glucose treatment groups (30 mM D-glucose concentration (Colombaioni et al., 2022)) for 24, 48, and 72 h, respectively.

(3) RNA extraction and qRT-PCR: After high-glucose treatment, total RNA was extracted from MPC5 cells using TRIzol reagent (Invitrogen, Waltham, MA, USA) following the manufacturer’s instructions. RNA concentration and purity were measured using a Nanodrop spectrophotometer (Thermo Fisher Scientific, Waltham, MA, USA). cDNA was synthesized using a reverse transcription kit (Biosharp, Hefei, China). qRT-PCR was performed to detect the relative mRNA expression levels of LTBP1 and ITGB6 using specific primers and ACTIN as the internal control. Reactions were conducted on an ABI 7500 Real-Time PCR System using SYBR Green PCR Master Mix (Yesen, China). Each experimental condition included three independent biological replicates, with each biological replicate measured in technical triplicate. The relative expression levels were calculated using the 2−ΔΔCt method, primer sequences were shown in Table 1.

Table 1 Primer sequences.

Primer sequence of ITGB6, LTBP1 and ACTIN.

Primers	Forward primer sequences	Reverse primer sequences	
ITGB6	CAACTATCGGCCAACTCATTGA	GCAGTTCTTCATAAGCGGAGAT	
LTBP1	CCAGTCCCAAGTCTCTTACCA	CTGGAAGCATCGGCCAAGT	
ACTIN	AGAGGGAAATCGTGCGTGAC	CAATAGTGATGACCTGGCCGT	

(4) Statistical analysis: All experimental data are presented as mean ± standard deviation. A two-way analysis of variance (two-way ANOVA) was performed to assess the main effects of treatment (Control vs. HG) and time (24, 48, 72 h), as well as their interaction. When a statistically significant interaction was observed, post hoc tests with Bonferroni correction were further conducted to compare groups at each time point. All statistical analyses were carried out using GraphPad Prism version 9.0, with P < 0.05 considered statistically significant.

Results

Differential gene expression and WGCNA analysis in the training set (GPL17586)

Differential gene expression analysis

To identify differentially expressed genes (DEGs) between DKD and control samples in the GPL17586 dataset, the “limma” package in R was used with thresholds of |log2FC| > 1 and adjusted P-value (P.adj) < 0.05. A total of 336 DEGs were identified, including 107 upregulated and 259 downregulated genes. A volcano plot was generated using the “ggpubr” package to visualize these results (Fig. 1A). The top 10 genes ranked by absolute log2FC values were selected for visualization in a heatmap created with the “ComplexHeatmap” package (Fig. 1B).

Figure 1 Differential expression analysis and WGCNA.

(A) Volcano plot visualizing differentially expressed genes (DEGs) in the GPL17586 dataset. Genes with |log2FC| > 1 and P.adj < 0.05 are highlighted in orange (upregulated, n = 107) and green (downregulated, n = 259). Gray points represent non-significant genes. (B) Heatmap displaying the expression patterns of the top 10 DEGs ranked by absolute log2FC values. The top panel shows the distribution of expression values, and the bottom panel presents the z-score normalized expression across samples. (C) Comparison of cuproptosis-related gene set enrichment scores (ssGSEA) between DKD and control groups. The y-axis represents ssGSEA scores, and the x-axis indicates sample groups. Statistical significance is denoted. (D) Sample hierarchical clustering based on Euclidean distance and cuproptosis gene set score. (E) Soft threshold selection for WGCNA network construction. Left panel shows the scale-free topology fit index (R2) for different soft thresholds (β). Right panel displays the mean connectivity across thresholds. The chosen threshold (β = 5) meets the scale-free criterion (R2 ≥ 0.85). (F) Co-expression module identification showing gene clustering and module color coding. (G) Heatmap of module-trait correlations between module eigengenes (MEs) and cuproptosis ssGSEA scores. Red indicates positive correlation, blue indicates negative correlation. The brown module shows the strongest positive correlation, and the blue module the strongest negative correlation. (H) Scatter plot of module membership (MM) vs. gene significance (GS) for the brown module. Genes with |MM| > 0.8 and |GS| > 0.6 (n = 390) were selected as key genes. (I) Scatter plot of module membership (MM) vs. gene significance (GS) for the blue module. Genes with |MM| > 0.8 and |GS| > 0.4 (n = 159) were selected as key genes. (**P < 0.01 vs. normal).

Cuproptosis gene set ssGSEA scores

The ssGSEA method was used to calculate the enrichment scores of the cuproptosis gene set for each sample. The results (Fig. 1C) showed the ssGSEA scores on the y-axis and sample groups on the x-axis, revealing significant differences in cuproptosis gene set enrichment scores between the DKD and control groups.

Hierarchical clustering of samples

The “WGCNA” package was used to perform weighted gene co-expression network analysis on the expression matrix of GPL17586 samples. Hierarchical clustering based on the Euclidean distance of gene expression was conducted to detect and exclude outlier samples. The clustering results (Fig. 1D) indicated good sample clustering, with no outliers requiring removal.

Soft threshold selection

To ensure the WGCNA network adhered to a scale-free topology, an optimal soft threshold (power) was determined from a range of 1 to 20. Two plots were generated to guide the selection.

The relationship between the soft threshold β and the scale-free topology fit index R2 (Fig. 1E, left).

The relationship between the soft threshold β and mean connectivity (Fig. 1E, right). A soft threshold β = 5 was selected based on R2 ≥ 0.85 and mean connectivity metrics.

Identification of co-expression modules

Using the selected soft threshold, a scale-free network was constructed, and genes were grouped into several co-expression modules, each assigned a unique color (Fig. 1F). A total of 7 co-expression modules (excluding the grey module) were identified.

Correlation analysis with cuproptosis gene set scores

The ssGSEA scores for the cuproptosis gene set were used as a phenotype, and the module eigengenes (ME) were correlated with these scores. A correlation heatmap (Fig. 1G) revealed that the brown module was most positively correlated, while the blue module was most negatively correlated with the ssGSEA scores.

Key module gene selection

For the brown and blue modules, the relationships between module membership (MM) and gene significance (GS) were analyzed. Scatter plots of MM vs. GS for the brown and blue modules were created using the “ggplot2” package (Figs. 1H and 1I).

Brown module: Genes with |MM| > 0.8 and |GS| > 0.6 were selected as key genes, resulting in 390 genes.

Blue module: Genes with |MM| > 0.8 and |GS| > 0.4 were selected, resulting in 159 genes.

Candidate gene screening

To identify differentially expressed genes associated with cuproptosis, we intersected the 366 differentially expressed genes with the key genes from the two co-expression modules, defining these intersections as candidate genes. Venn diagrams were generated using the R package “VennDiagram” to visualize the results. The brown module contained 124 differentially expressed genes (Fig. 2A), while the blue module contained 28 differentially expressed genes (Fig. 2B).

Figure 2 Screening of candidate key genes related to cuproptosis in DKD.

(A–B) Venn diagram showing the intersection between differentially expressed genes (DEGs) and key genes from the brown (A) and blue (B) co-expression modules, identifying overlapping candidate genes. (C–D) Feature importance rankings of candidate genes from the brown (C) and blue (D) modules identified using the XGBoost algorithm on the GPL17586 training set. (E–F) LASSO regression analyses for the brown (E) and blue (F) modules. Left: coefficient profile of genes against log(λ); Right: cross-validation curve showing deviance vs. log(λ). (G–H) Venn diagram illustrating the intersection of candidate genes from the brown (G) and blue (H) modules identified by both XGBoost and LASSO algorithms, resulting in final key genes.

Key gene selection using machine learning

Two machine learning algorithms were applied to screen the differentially expressed key genes in the brown and blue modules, respectively. The selected candidate genes were subsequently validated for expression.

Xgboost selection

Using the GPL17586 dataset as the training set, we applied the XGBoost algorithm to the 124 differentially expressed genes in the brown module, identifying 13 candidate genes, which were ranked by importance (Fig. 2C). For the 28 differentially expressed genes in the blue module, 18 candidate genes were identified, and similarly ranked by importance (Fig. 2D).

LASSO selection

The GPL17586 dataset was also used as the training set for LASSO regression analysis with the R package “glmnet.” In the brown module, the minimum Lambda value was 0.0039, corresponding to the smallest residual sum of squares, and 12 candidate genes were selected (Fig. 2E, left). These genes include DAO, DCXR, ETFB, FMO1, FMO4, HSD17B14, KIAA1191, MAOA, MORN2, PCBD1, PCK1, and SLC34A1.

For the blue module, the minimum Lambda value was 0.0083, resulting in the selection of 10 candidate genes (Fig. 2F), including CCL2, CCND2, CLDN1, FBN1, INHBA, ITGB6, LTBP1, SFRP2, SLIT3, and TPM1.

Intersection screening

By taking the intersection of candidate genes identified by both machine learning algorithms in the brown module, we identified seven key genes (Fig. 2G): ETFB, HSD17B14, KIAA1191, MAOA, MORN2, PCK1, and SLC34A1. For the blue module, the intersection resulted in nine key genes (Fig. 2H): CCL2, CCND2, CLDN1, FBN1, INHBA, ITGB6, LTBP1, SLIT3, and TPM1.

Expression validation

To validate the expression of the key genes identified by machine learning, we compared the differential expression of candidate genes between DKD samples and control samples in both the training set (GPL17586) and the validation set (GPL571) using the Wilcoxon test (P < 0.05). The results showed significant differences in the expression of the candidate genes between DKD and control samples. The analysis results were visualized using box plots created with the R package “ggplot2.” Genes that showed significant differential expression (P < 0.05) and consistent trends in both the training and validation sets were defined as key genes.

As shown in Fig. 3A, the differential expression analysis of the key genes in the brown module from the training set (GPL17586) revealed significant differences for all genes. In contrast, the analysis in the validation set (GPL571) (Fig. 3B) showed no significant differences, and no expression was observed for KIAA1191 and MORN2. Figure 3C presents the differential expression analysis of the blue module in the training set (GPL17586), where all genes showed significant differences. Figure 3D presents the results from the validation set (GPL571), where significant expression differences were observed for ITGB6 and LTBP1, both of which demonstrated consistent upregulation in both the training and validation sets. Together, these results confirm ITGB6 and LTBP1 as key genes associated with DKD.

Figure 3 Expression validation of candidate key genes in training and validation datasets.

(A–D) Expression differences of key genes in the brown (A–B) and blue (C–D) modules between DKD and control samples in both training and validation sets. Disease group (red) and control group (green) are shown in box plots. (*P < 0.05, ***P < 0.005, ****P < 0.0001 vs. normal).

ROC analysis

To validate the diagnostic performance of the key genes for DKD, we conducted ROC curve analysis using the R package “pROC” in both the training set (GPL17586) and the validation set (GPL571), and calculated the area under the curve (AUC). As shown in Figs. 4A–4D, the AUC values for ITGB6 and LTBP1 in both the training and validation sets were greater than 0.7, indicating that both genes exhibit good diagnostic performance for DKD.

Figure 4 ROC and nomogram of key genes.

(A–D): ROC analysis of ITGB6 and LTBP1 genes in both training and validation sets, with AUC > 0.7, confirming consistent diagnostic value. (E) Nomogram constructed using ITGB6 and LTBP1 based on the training set (GPL17586). (F) Calibration curve of the nomogram model showing predicted vs. actual disease risk. (G) Decision curve analysis (DCA) of the diagnostic model. The blue curve (nomogram model) lies above the gray line (treat-all) and the black line (treat-none), indicating a positive net benefit across a wide range of threshold probabilities. (H) ROC analysis of the nomogram-based diagnostic model in the training set (GPL17586) for DKD.

To further assess the impact of the key genes on the clinical probability of DKD, we constructed a nomogram based on all samples from the training set (GPL17586) using the R package “rms.” First, scores were assigned based on the diagnostic performance of each key gene for DKD, and the total score (Total Point) was calculated. The probability of DKD occurence was then inferred from the total score. A higher total score corresponds to a higher probability of disease occurrence. As shown in Fig. 4E, the model constructed using ITGB6 and LTBP1 had a total score of 113, resulting in a DKD prediction probability of 0.959, thereby demonstrating strong diagnostic potential.

The predictive ability of the nomogram model was further evaluated using calibration curves, decision curves (DCA), and ROC analysis. The calibration curve was assessed using the Hosmer-Lemeshow (HL) test. A P-value greater than 0.05 and a slope of the calibration curve close to 1 indicate good model fit. As shown in Fig. 4F, the P-value was 0.583, and the slope of the calibration curve was close to 1, suggesting that the model fits well.

Figure 4G presents the DCA decision curve, where the diagnostic model curve (blue) is above the gray baseline, and the net benefit (Net Benefit) is greater than 0, confirming that the model performs well in predicting DKD.

Finally, we assessed the predictive ability of the nomogram model in the training set using ROC analysis with the R package “pROC.” As shown in Fig. 4H, the AUC value was 0.89, demonstrating that the nomogram model possesses strong predictive capability.

Immune infiltration analysis

To explore the relationship between the key genes ITGB6 and LTBP1 and the immune microenvironment, we used the R package “GSVA” to calculate the enrichment scores of 28 immune cell types in DKD samples and control samples using the ssGSEA algorithm, based on the training set (GPL17586). As shown in Fig. 5A, a stacked bar plot was created to illustrate the proportion of immune cell infiltration between the two groups, with different colors representing different cell types, and the y-axis indicating the relative proportion of immune cells for each sample.

Figure 5 Immune infiltration and functional enrichment analysis of key genes in DKD.

(A) Stacked bar plot illustrating the relative proportions of 28 immune cell types in DKD and control samples from the GPL17586 dataset, as calculated by ssGSEA. The y-axis represents the cumulative enrichment score per sample. (B) Violin plot showing the infiltration levels of 17 immune cell types that exhibited significant differences between DKD and control groups (C) Spearman correlation heatmap between key genes (ITGB6 and LTBP1) and differentially infiltrated immune cells. Red indicates positive correlation, blue indicates negative correlation. (D) KEGG pathway analysis for top 20 significant pathways for the 152 differentially expressed genes from the Brown and Blue modules. (E) GO analysis with enriched pathways shown in circular diagram with descriptions. (F) PPI network analysis of the 28 candidate genes from the Blue module with node degree values. (*P < 0.05, **P < 0.01, ***P < 0.005, ****P < 0.0001 vs. normal).

We applied the Wilcoxon test to compare immune cell infiltration between the two groups. As shown in Fig. 5B, 17 immune cell types exhibited significant differences (P < 0.05) between the groups, including activated B cells, CD56bright natural killer cells, central memory CD4 T cells, effector memory CD4 T cells, etc. To further investigate the relationship between ITGB6 and LTBP1 and these differential immune cells, Spearman correlation analysis was conducted using the R package “psych.” The results (Fig. 5C) indicated that ITGB6 and LTBP1 were significantly positively correlated with Activated B cells and Central memory CD4 T cells, and significantly negatively correlated with Neutrophils.

Enrichment analysis

We selected 152 differential genes from the Brown and Blue modules and performed GO and KEGG pathway enrichment analysis using the “clusterProfiler” package. Based on a P-value threshold of < 0.05, 548 GO biological functions were enriched, including 399 biological processes (BP), 117 molecular functions (MF), and 32 cellular components (CC), as shown in Fig. 5E. Notable biological processes, such as small-molecule degradation and amino acid metabolism, showed significant changes. The KEGG analysis identified 52 enriched pathways, with significant alterations in pathways such as β-alanine metabolism and butyrate metabolism, as shown in Fig. 5D.

Protein-protein interaction analysis

To investigate the interactions between ITGB6 and LTBP1 within the key genes of the Blue module, we constructed a protein-protein interaction (PPI) network for the 28 candidate genes using the STRING database (https://string-db.org), and visualized the network using Cytoscape software. As shown in Fig. 5F, the PPI network contains 25 nodes and 162 edges, with a confidence score of 0.4. The most strongly associated gene in the network is fibronectin 1 (FN1), which is a key gene related to fibrosis.

Key gene and related protein analysis

The PANTHER classification system is a comprehensive and annotated gene family database that categorizes proteins and their corresponding genes. To further investigate the classification of the key genes at the protein level, we utilized this database to analyze the key genes and obtain classification information for ITGB6 and LTBP1. The results are shown in Table 2.

Table 2 Classification of proteins encoded by key genes.

The annotation of proteins encoded by key genes identified in the study. According to the PANTHER classification system, LTBP1 (latent transforming growth factor beta binding protein 1) is categorized as an extracellular matrix structural protein, whereas ITGB6 (integrin beta 6) is classified as an integrin family member. Gene identifiers (HGNC) and UniProt accession numbers are provided for reference.

Mapped IDs	Gene ID	PANTHER Family/Subfamily	Hide column
PANTHER protein class	
LTBP1	HUMAN|HGNC=6714|UniProtKB=Q14766	LATENT-TRANSFORMING GROWTH FACTOR BETA-BINDING PROTEIN 1 (PTHR24034:SF140)	Extracellular matrix structural protein	
ITGB6	HUMAN|HGNC=6161|UniProtKB=P18564	INTEGRIN BETA-6 (PTHR10082:SF11)	Integrin	

Expression of ITGB6 and LTBP1 in renal podocytes under high glucose culture

To investigate the impact of a high-glucose environment on the expression of LTBP1 and ITGB6 in MPC5 cells, we exposed MPC5 cells to a high-glucose condition of 30 mM for 24, 48, and 72 h. Subsequently, we measured the relative mRNA expression levels of LTBP1 and ITGB6 using qRT-PCR.

As shown in Fig. 6, the expression of both LTBP1 and ITGB6 significantly increased with prolonged high-glucose treatment. For LTBP1, the relative expression levels at 48 h were significantly higher than the control group, with a P-value of 0.0012. Similarly, ITGB6 exhibited a significant time-dependent upregulation, with relative expression levels at 24, 48, and 72 h showing P-values of 0.035, 0.0014, and 0.0034, all significantly higher than the control group.

Figure 6 Relative mRNA expression of LTBP1 and ITGB6 in MPC5 cells.

MPC5 cells were treated with high glucose (30 mM) for 24, 48, and 72 h. (A) LTBP1 expression increased significantly compared to control (P < 0.05) at 48 h. Data are shown as mean ± SEM, n = 3. *P < 0.05. (B) ITGB6 expression increased significantly compared to control (P < 0.05) at 24, 48 and 72 h. Data are shown as mean ± SEM, n = 3. *P < 0.05; **P < 0.01, ***P < 0.001.

Discussion

In this study, we investigated the potential involvement of cuproptosis-related mechanisms in the pathogenesis of diabetic kidney disease (DKD). Although previous studies have examined the relationship between copper homeostasis and metabolic diseases or chronic kidney disease (Jia et al., 2019; Chen, Li & Liu, 2023), research specifically addressing the role of copper in DKD remains limited. Our findings demonstrate a significant upregulation of two cuproptosis-related genes, ITGB6 and LTBP1, in glomerular tissues from DKD patients. Furthermore, the expression levels of these genes were positively associated with disease progression, suggesting that ITGB6 and LTBP1 may serve as key mediators linking cuproptosis to the DKD immune microenvironment.

Both ITGB6 and LTBP1 are extracellular matrix (ECM) regulatory proteins that mediate epithelial–stromal interactions and fibrosis via TGF-β activation (Lorda-Diez et al., 2010; Bekhouche et al., 2016; Lanktree et al., 2023; Voinescu et al., 2024). Previous studies have shown that ITGB6 overexpression aggravates glomerular injury and interstitial fibrosis in DKD, highlighting its important role in disease progression (Qi et al., 2010; Yao et al., 2024). In our study, differential expression analysis confirmed significant upregulation of ITGB6 and LTBP1 in DKD patients and in high-glucose–treated MPC5 podocytes, consistent with their established roles in ECM remodeling and fibrotic signaling (Ducceschi et al., 2014).

Podocytes are crucial components of the glomerular filtration barrier, and their damage represents a key pathological event in early DKD, contributing to proteinuria and glomerulosclerosis (Liu et al., 2025). High-glucose stimulation in vitro mimics the hyperglycemic milieu of DKD, inducing oxidative stress, inflammation, apoptosis, and autophagy in podocytes (Zhao et al., 2022). In our experiments, the expression of ITGB6 and LTBP1 increased in a time-dependent manner under high-glucose conditions, reinforcing their potential roles in hyperglycemia-induced injury.

Disrupted copper metabolism has been implicated in DKD pathogenesis (Hamasaki, Kawashima & Yanai, 2016; Chen et al., 2024). Our results extend this knowledge by linking ITGB6 and LTBP1 expression to cuproptosis-associated pathways (Li, 2020; Takizawa et al., 2023). Immune infiltration analysis revealed positive correlations between ITGB6/LTBP1 expression and elevated proportions of activated B cells and CD4+T cells in DKD tissues (Zhou et al., 2023). These immune subsets drive chronic inflammation and renal injury, suggesting that ITGB6 and LTBP1 may promote immune cell recruitment or activation rather than exerting protective effects.

Using WGCNA, we identified ITGB6 and LTBP1 as central hub genes in the module most strongly associated with cuproptosis-related gene expression. Functional enrichment analysis of this module highlighted pathways involving mitochondrial function, metal ion homeostasis, and oxidative stress—key features of cuproptosis. Mechanistically, ITGB6 activates latent TGF-β, whereas LTBP1 stabilizes TGF-β complexes and regulates extracellular matrix (ECM) organization (Feng et al., 2023; Yao et al., 2024; Zhu et al., 2024). Their co-expression with cuproptosis-related genes suggests potential roles as upstream modulators or downstream effectors of copper-induced metabolic stress.

Moreover, machine learning and ROC analyses demonstrated strong diagnostic potential for these genes, with both ITGB6 and LTBP1 achieving AUC > 0.7 in training and validation cohorts, indicating robust discriminatory power for DKD diagnosis and progression monitoring. Together, these findings support ITGB6 and LTBP1 as not only potential biomarkers but also as key candidates for further mechanistic studies in the context of cuproptosis and DKD.

Limitations

This study indicates that cuproptosis-related pathways may contribute to DKD pathogenesis, with significant upregulation of ITGB6 and LTBP1 observed in DKD and their potential involvement in modulating immune microenvironment. Although our study has yielded significant findings, there are still several aspects that warrant further improvement. First, the absence of a mannitol control group limits our ability to exclude osmotic effects. Second, classical cuproptosis signaling pathways—including FDX1, LIAS, and DLAT protein lipoylation, as well as mitochondrial membrane potential alterations—require further investigation to better substantiate the link between cuproptosis and DKD. Finally, additional studies employing in vitro and in vivo DKD models, combined with Western blotting, immunostaining, and functional inhibition assays targeting ITGB6 and LTBP1, are necessary to elucidate their roles in cuproptosis and inflammatory signaling under hyperglycemic conditions.

Supplemental Information

Supplemental Information 1 Proposed model of ITGB6 and LTBP1 involvement in DKD progression.

High glucose induces podocyte activation and upregulation of ITGB6 and LTBP1, which correlate with immune cell infiltration and cuproptosis, contributing to diabetic kidney disease (DKD) progression.

Supplemental Information 2 Original qPCR data.

Raw expression data of the cuproptosis-related genes ITGB6 and LTBP1 in MPC5 cells cultured under 30 mM glucose condition

Supplemental Information 3 Original Images and Data.

Supplemental Information 4 MIQE checklist.

We thank the members of our laboratory for their technical support and helpful discussions. We also appreciate the assistance of the core facilities at Wuhu Hospital of Traditional Chinese Medicine.

Abbreviations

DKD Diabetic Kidney Disease

ITGB6 Integrin Beta-6

LTBP1 Latent Transforming Growth Factor Beta Binding Protein 1

MPC5 Mouse Podocyte Cell Line

ROC Receiver Operating Characteristic

AUC Area Under the Curve

ssGSEA Single-Sample Gene Set Enrichment Analysis

WGCNA Weighted Gene Co-Expression Network Analysis

PPI Protein-Protein Interaction

Additional Information and Declarations

Competing Interests

The authors declare that they have no competing interests.

Author Contributions

Suying Hu performed the experiments, prepared figures and/or tables, and approved the final draft.

Mengdi Tian performed the experiments, analyzed the data, prepared figures and/or tables, and approved the final draft.

Wenjia Hu performed the experiments, prepared figures and/or tables, and approved the final draft.

Liang Yao performed the experiments, prepared figures and/or tables, and approved the final draft.

Ying Tang analyzed the data, prepared figures and/or tables, and approved the final draft.

Wei Shen analyzed the data, authored or reviewed drafts of the article, and approved the final draft.

Qing He analyzed the data, authored or reviewed drafts of the article, and approved the final draft.

Jing Xu analyzed the data, authored or reviewed drafts of the article, and approved the final draft.

Huan Yao conceived and designed the experiments, performed the experiments, analyzed the data, prepared figures and/or tables, and approved the final draft.

Lei Ji conceived and designed the experiments, prepared figures and/or tables, and approved the final draft.

Feifei Fan conceived and designed the experiments, prepared figures and/or tables, and approved the final draft.

Shiqiang Liu conceived and designed the experiments, performed the experiments, prepared figures and/or tables, authored or reviewed drafts of the article, and approved the final draft.

Zhen Wang conceived and designed the experiments, authored or reviewed drafts of the article, and approved the final draft.

Data Availability

The following information was supplied regarding data availability:

The raw data are available in the Supplemental Files.

The data is available at GEO: GPL17586 and GPL571.

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
