# Peer review of "The cuproptosis-related gene ITGB6 and LTBP1 may be associated with diabetic kidney disease progression and immune cell infiltration"

_PeerJ, doi:10.7717/peerj.20346_

## Round 0.1 · original submission · Minor Revisions

· Academic Editor

Minor Revisions

Two experts reviewed your manuscript. Since both recommend a minor revision, I have decided that you need a minor revision. However, you will need much effort for its revision, almost equal to a major revision. Although it will be difficult for you to add experimental verification, please read their comments carefully and follow their advice wherever possible; explain the reason why otherwise.

·

Basic reporting

The cuproptosis-related gene ITGB6 and LTBP1 may be associated with DKD progression and immune cell infiltration.

The manuscript explored a relevant and significant subject: looking into the possible significance of cuproptosis-related genes in diabetic kidney disease (DKD) and how they are linked to immune cell infiltration. In general, the study is well-constructed, with a clear procedure and efficient use of bioinformatic analysis and machine learning-based gene selection. Figures, tables, and raw data are all suitable. However, the study is still only correlational; even while the results are interesting, we can't be sure what caused them without doing functional experiments like gene knockdown or overexpression. Additionally, there are a couple of points that need further attention:

1. The authors used extremely high glucose concentrations during the in vitro experiment (45 mM) that may never occur in a physiological hyperglycemia setting. This needs further justification of why they chose this level, although they touched on this inadequately in the discussion.

2. Although the manuscript is mostly written in clear and professional English, the discussion section requires more wording revisions for better readability.

Experimental design

Clear experimental design.

Validity of the findings

The use of an appropriate statistical analysis tool, the use of 2 independent datasets, and the in vitro validation all support the findings and ca use and lead to a robust conclusion.

Reviewer 2 ·

Basic reporting

The manuscript would benefit from revision for grammatical accuracy and improved consistency in technical terminology, particularly in the use of terms such as “cuproptosis.” The literature is generally well-cited, with appropriate and recent references; however, the connection between cuproptosis and diabetic kidney disease (DKD) requires deeper contextualization. Including additional references that explore cuproptosis-related mechanisms in renal pathophysiology would strengthen the background.
The rationale for dataset selection should be made explicit. A recommended addition is a statement justifying the use of GSE96804 as the training set (larger sample size, better signal-to-noise ratio) and GSE30122 as the validation set (independent cohort, different platform).

The figure legends, especially for multi-panel figures, are brief and would benefit from more descriptive captions to enhance interpretability.

The manuscript presents a coherent progression from bioinformatics analysis to experimental validation. However, the proposed role of cuproptosis is based solely on gene expression data, without direct experimental evidence (e.g., lipoylation, Fe–S cluster disruption, or mitochondrial stress markers). This limits the strength of the mechanistic conclusions and should be addressed in the discussion.

Experimental design

The manuscript addresses an important knowledge gap concerning the potential role of copper-dependent cell death (cuproptosis) in renal pathology. However, this gap should be more explicitly defined in the introduction, with a clearer rationale for focusing on cuproptosis in podocytes or renal tissue to improve the framing of the study’s biological relevance.

The in vitro model using high-glucose treatment lacks an essential osmotic control (e.g., mannitol-treated group), which limits the ability to attribute observed gene expression changes specifically to hyperglycemia. This methodological limitation should be addressed either experimentally or through a more explicit discussion of its absence in the manuscript.

A central concern is the absence of direct assessment of cuproptosis. While the study is framed around this mechanism, no biochemical markers (e.g., lipoylation status, Fe–S cluster integrity, mitochondrial stress indicators) were measured in the cell model. This renders the proposed mechanistic link speculative and weakens the interpretation of gene expression results in terms of cuproptosis.

Additionally, some reporting omissions should be corrected to meet transparency and reproducibility standards:
Research Resource Identifiers (RRIDs) for the MPC5 cell line are not provided.
A statement on ethical approval or compliance for the cell experiments is missing.
The feature selection criteria in Lasso and XGBoost (e.g., hyperparameters, validation strategy) are insufficiently described.
The cut-off values used for identifying DEGs, constructing WGCNA modules, and generating ROC curves should be clearly reported.
There is no discussion of batch effect correction, which is particularly relevant given the use of datasets from different platforms.
It is unclear whether the qRT-PCR replicates represent biological or technical replicates — this distinction should be clarified.
The rationale for dataset selection should be briefly explained.

Validity of the findings

Reframe conclusions to reflect the correlative nature of the gene expression findings and avoid overstatement regarding mechanistic involvement in cuproptosis.

Clarify whether statistical tests for qRT-PCR were based on biological or technical replicates.
Explicitly discuss batch correction or normalization procedures and justify the use of cross-platform datasets.

If feasible, consider incorporating or proposing direct experimental assessment of cuproptosis features in the cell model.

---

## Round 0.2 · accepted · Accept

· Academic Editor

Accept

As the handling editor, I confirmed that the authors addressed all points raised by the reviewers. Since both recommended minor revisions previously, I did not bother them to review the revised manuscript. Now I recommend its acceptance to the section editor. Congratulations!